# Microbial Profile and Antibiotic Resistance Patterns in Bile Aspirates from Patients with Acute Cholangitis: A Multicenter International Study

**DOI:** 10.3390/antibiotics14070679

**Published:** 2025-07-04

**Authors:** Matei-Alexandru Cozma, Mihnea-Alexandru Găman, Camelia Cristina Diaconu, Arthur Berger, Frank Zerbib, Radu Bogdan Mateescu

**Affiliations:** 1Faculty of Medicine, “Carol Davila” University of Medicine and Pharmacy, 050474 Bucharest, Romania; radu.mateescu@umfcd.ro; 2Department of Gastroenterology and Hepatology, Colentina Clinical Hospital, 020125 Bucharest, Romania; 3Department of Hematology, Center for Clinical and Basic Research (CCBR Clinic), 030463 Bucharest, Romania; mihneagaman@yahoo.com; 4Department of Cellular and Molecular Pathology, Stefan S. Nicolau Institute of Virology, Romanian Academy, 030304 Bucharest, Romania; 5Department of Internal Medicine, Clinical Emergency Hospital of Bucharest, 14461 Bucharest, Romania; 6Haut-Lévêque Hospital, Centre Hospitalier Universitaire de Bordeaux, 33600 Pessac, France; arthur.berger@chu-bordeaux.fr (A.B.); frank.zerbib@chu-bordeaux.fr (F.Z.)

**Keywords:** acute cholangitis, antimicrobial resistance, bile cultures, multidrug-resistant bacteria

## Abstract

Objectives: Significant differences in antibiotic resistance (AR) rates and multi-drug resistant (MDR) bacteria incidence exist in patients with acute cholangitis (AC) from different countries or regions. We aim to characterize and compare the microbial spectrum and AR patterns in patients with AC from two tertiary centers in Europe. Methods: We conducted a prospective, observational, multicentric study including patients diagnosed with AC and a positive bile culture, admitted to the Colentina Clinical Hospital (CCH), Bucharest, Romania, and the Haut-Lévêque Hospital (HLH), Bordeaux, France, between April 2022 and October 2023. Results: We included a total of 144 patients from the CCH with 190 positive bile cultures (31 patients had up to five episodes of AC during the study period) and 241 identified microbial strains, and 62 patients from the HLH with 67 positive bile cultures (5 patients had two episodes of AC) and 194 identified microbial strains. The most frequently isolated bacteria were *Escherichia coli* (30.70%) and *Pseudomonas* spp. (27.80%) in the CCH group, and *Enterococcus faecalis* (15.46%) and *Escherichia coli* (22/11.34%) in the HLH group. Furthermore, 51 (21.16%) of the strains identified in the CCH group and 15 (7.21%) in the HLH group were MDR, such as extended-spectrum beta-lactamase-producing *Enterobacteriaceae* or carbapenemase-producing *Enterobacterales*. The resistance rates for common antibiotics were 13.69% in the CCH group vs. 8.76% in the HLH group for ceftriaxone, 9.54% vs. 2.06% for meropenem, 16.59% vs. 6.70% for piperacillin/tazobactam, and 25.31% vs. 7.73% for levofloxacin. Conclusions: This comparative study shows significant differences between these countries in terms of the AR rates and MDR bacteria prevalence, highlighting the role of bile cultures as a safe and cost-effective method for guiding antibiotic treatment, thereby reducing the AR rates and complications.

## 1. Introduction

Acute cholangitis (AC) represents a life-threatening infectious disease of the biliary tract, with increased risk for poor prognosis in the absence of swift treatment [1,2,3]. Prevalence has been rising globally to 8–12 cases/100,000 people, mainly due to the growing burden of gallstone disease and the increasing incidence of malignant stenoses of the common bile duct (CBD) [3]. Systemic infection occurs in up to 20% of cases, and the mortality rate remains at around 10%, despite a significant decline in recent years through widespread implementation of endoscopic interventions [4,5].

AC appears in the setting of impaired bile flow caused by benign (gallstones, benign ampuloma) or malignant (pancreatic head cancer or cholangiocarcinoma) CBD obstructions. Elevated CBD pressure and retrograde flow of the bile lead to bacterial proliferation and eventually translocation into the systemic circulation, causing sepsis and multiple organ dysfunction [6,7].

Treatment must be initiated as soon as possible, and consists of prompt and adequate hemodynamic resuscitation, wide-spectrum antibiotic therapy, and early interventional restoration of the bile flow [8,9,10,11]. Traditionally, microbial agents were isolated by blood cultures; however, these are positive in only 21% to 71% of the samples [1,12]. Because of this, bile sample aspiration for microbiological analysis during endoscopic retrograde cholangiopancreatography (ERCP) is now increasingly used to guide initial empiric treatment, being recognized as a cheap, fast, and more specific method, with up to 100% positivity rates [1,13,14,15].

Antimicrobial resistance (AMR) is a significant cause of morbidity, mortality, and unnecessarily high medical costs globally, responsible for over 1 million deaths each year [16,17,18]. AC makes no exception, as the increasing prevalence of MDR bacteria has led to increased mortality and complication rates [19,20]. Although AMR is a global major health problem, significant variations exist at the national or even regional level, with high consumption of broad-spectrum antibiotics and several socio-economic aspects (inefficient infection control, population density, and national tourism profile) being the most relevant risk factors (RFs) [21,22]. Unfortunately, to this date, only limited comparative data exist on epidemiological aspects of AC patients in Romania.

While previous studies exist and have provided some limited information on the microbiological spectrum and resistance patterns in AC patients from both countries, Romania and France, they are often unicentric, include small cohorts, and are limited to a region. Our study focuses on the knowledge gap regarding the comparative findings between multiple regions by comparing similar data from two regions with distinct antibiotic policies, infection control strategies, and healthcare infrastructure in Eastern (Romania) and Western Europe (France). Additionally, while the importance of blood cultures in systemic infections has been thoroughly studied and validated, we chose to focus on bile cultures in order to increase the clinical value of our findings, considering their increased specificity and the limited attention they have received. Last, but not least, we explored the prevalence, impact, and risk factors associated with MDR bacterial infection and, at the same time, investigated the impact of the prior endoscopic interventions’ resistance patterns, thus moving beyond standard descriptive epidemiological analysis, and thereby increasing the relevance and impact of our results.

The main objective of this study was to analyze and compare the microbial spectrum and the AR patterns of pathogens isolated from bile cultures collected from patients with AC in two referral centers from Eastern and Western Europe. Additional objectives were to identify the incidence of MDR bacteria among etiological agents and the RFs associated with such infections and, at the same time, to analyze the influence of previous endoscopic interventions on resistance patterns and MDR bacteria incidence.

## 2. Results

### 2.1. Patients and General Characteristics

Of the 823 patients admitted with a positive diagnosis of AC in both medical units, 205 patients met the inclusion criteria and were included in this study (143 from the CCH and 62 from the HLH). The inclusion flowchart is shown in Figure 1, below. Collectively, these patients had 257 positive bile samples processed (190 from the CCH and 67 from the HLH), leading to the identification and analysis of 436 etiological agents (241 in the CCH group and 194 in the HLH group). Notably, 32 patients from the CCH group and 5 patients from the HLH group had bile samples collected during multiple hospitalizations within the study period. Additionally, a single patient from the CCH group presented with two distinct positive bile cultures during the same hospitalization.

The mean age was 68.49 years old (SD = 10.40) for the CCH group, and 70.34 years old (SD = 9.02) for the HLH group. A non-significant male predominance was observed in both study groups: 83 patients (58.04%) in the CCH group and 40 patients (64.51%) in the HLH group. A total of 83 patients (58.04%) from the CCH group and 47 patients (75.80%) from the HLH group experienced an episode of AC caused by a malignant etiology, with no significant difference between the two groups. In terms of endoscopic history, a significantly smaller proportion of patients from the CCH group had naïve papilla compared to the HLH group (46.15% vs. 66.13%, *p* = 0.0098).

Of the 189 and 67 episodes of AC that occurred during the study period in both hospitals, a significantly higher proportion occurred in patients with a biliary stent already in place for the CCH group (103 cases, 54.5%), as opposed to the HLH group (25 cases, 37.31%) (*p* = 0.0225). The mean length of hospitalization was longer for the CCH group, at 5.64 days, compared to 3.35 days for the HLH group (*p* < 0.0001). Complications, such as post-ERCP pancreatitis or post-sphincterotomy bleeding, occurred in 57 (30.15%) episodes from the CCH group and in 13 (19.40%) from the HLH group (*p* = 0.1107). Six (3.17%) and four (5.97%) AC episodes resulted in admission to the ICU department (*p* = 0.2941).

A total of 44 cases of AC for the CCH group occurred in patients who had had a different bile sample collected during a previous episode of AC, but with a negative result. Unfortunately, we did not have access to such data for the HLH group. A total of 37 blood cultures were collected from the CCH group patients hospitalized, of which only 19 turned out to a positive; for the HLH group, 31 blood cultures were collected, of which 22 were positive. Higher values were detected for total bilirubin, white blood cells, and C-reactive protein levels in the subgroup of patients with a stented CBD, but with no significant difference. All clinical and paraclinical characteristics of the patients included in this study are detailed in Table 1.

### 2.2. Etiologies of Acute Cholangitis

Malignant stenosis was the most common etiology of AC overall (58.04% and 75.80%), as well as within the subgroups of patients with biliary stents (77.92% and 76.19%). After performing a comparative analysis using the chi-squared test, we observed a significant difference between the groups of patients with naive papilla (*p* < 0.001), between the subgroups of patients from the CCH (*p* < 0.001), and between the total number of patients (*p* = 0.017) from the two medical units.

The most common etiology of AC in the CCH group was represented by choledocholithiasis (32.17%), followed by hilar cholangiocarcinoma (24.48%) and pancreatic head cancer (17.48%). Other etiologies were represented by distal cholangiocarcinoma and extrinsic compression (liver or lymph node metastases, chronic pancreatitis, or Mirizzi syndrome −6.29%). In the HLH patient group, the most common etiologies were pancreatic head cancer (41.46%), distal cholangiocarcinoma (14.52%), and hilar cholangiocarcinoma (11.29%). Other significant etiologies included malignant ampuloma (9.67%) and extrinsic compression (8.05%) (illustrated in Figure 2).

### 2.3. Microbial Agents Associated with AC

The majority of positive bile cultures were taken from patients with a stented CBD in both medical units (*p* = 0.0027). A significant difference between the two groups was in terms of the number of bacterial strains identified in each culture. While in the CCH group, the vast majority of cultures were monomicrobial (142 cases, 74.73%), in the HLH group, polymicrobial cultures (at least four distinct bacterial isolates) were more common (18 cases, 26.86%), with no such cases recorded in the CCH group. A detailed classification is shown in Table 2.

Gram-negative microorganisms represented the vast majority of bacteria identified in both study groups (detailed in Table 3 and illustrated in Figure 3). *Escherichia coli* (74, 30.70%), *Pseudomonas* spp. (68, 28.21%), *Klebsiella* spp. (40, 16.59%), and *Enterococcus* spp. (37, 15.35%) were the most common pathogens identified in the CCH study group. At the same time, less commonly identified species in this group included *Achromobacter denitrificans* (n = 1), *Cronobacter sakazakii* (n = 1), *Raoultella ornithinolytica* (n = 1), and *Proteus mirabilis* (n = 3). In the HLH study group, the most frequently identified bacteria were *Enterococcus* spp. (59, 30.41%), *Escherichia coli* (22, 11.34%), and *Streptococcus* spp. (21, 10.82%). Less frequently identified microbial strains included *Candida* spp. (n = 16), *Clostridium perfringens* (n = 3), *Serratia marcescens* (n = 5), and *Veillonella parvula* (n = 1).

In the comparative analysis using Fisher’s exact test, we observed significant differences when comparing all three subgroups from both medical units (“Naive Papilla” CCH vs. HLH, “Stented CBD” CCH vs. HLH, and “Total” CCH vs. HLH) for *Enterococcus* spp. (more frequent in the HLH group; *p* = 0.0112, *p* = 0.0122 and *p* = 0.0002), *Escherichia* spp. (more frequent in the CCH group; *p* = 0.0143, *p* = 0.0001 and *p* < 0.0001) and for *Pseudomonas* spp. (also more frequent in the CCH group; *p* < 0.0001 for all three subgroup comparisons). A further comparative analysis between the total number of patients can also be seen in Table 3 and Figure 3.

### 2.4. Resistance Patterns

As depicted in Table 4 and Figure 4, the resistance rates to most antibiotics were higher in the CCH group compared to the HLH group, with few exceptions seen for metronidazole, piperacillin/tazobactam, amoxicillin/clavulanic acid, and colistin. Moreover, a comparative statistical analysis using Fisher’s exact test demonstrated significant differences for several antibiotics. For example, significant resistance differences were observed for ceftriaxone (*p* = 0.03), ciprofloxacin (*p* < 0.0001), levofloxacin (*p* < 0.0001), meropenem (*p* = 0.0003), and cefazolin (*p* < 0.0001), to which the majority of the bacteria in the CCH group showed resistance. Conversely, significantly higher resistance was noted in the HLH group for piperacillin/tazobactam (*p* < 0.0001) and amoxicillin/clavulanic acid (*p* < 0.0001).

Table 5 and Figure 5 further detail the AR rates for the etiologic agents in the two groups, stratified by endoscopic history. Data shows that, in both groups, resistance rates were higher in the patients with biliary stents, with exceptions such as colistin, linezolid, and cefotaxime for the CCH group, and ceftriaxone, meropenem, and amoxicillin/clavulanic acid for the HLH group. Significant inter-subgroup differences were identified in the CCH group for ciprofloxacin (*p* = 0.045), levofloxacin (*p* = 0.012), cefazolin (*p* = 0.042), and linezolid (*p* = 0.034). However, no significant differences were detected among the HLH subgroups.

Furthermore, we opted for a comparative analysis of the same subgroups across the two hospitals (Naive Papilla–CCH vs. Naive Papilla–HLH and Stented CBD–CCH vs. Stented CBD–HLH). The comparison of the “Naive Papilla” subgroups revealed significant differences for piperacillin/tazobactam (*p* = 0.022), ciprofloxacin (*p* = 0.002), and levofloxacin (*p* = 0.043). For the “stented CBD” subgroups, significant differences were noted for piperacillin/tazobactam (*p* = 0.0059), ciprofloxacin (*p* = 0.0056), levofloxacin (*p* = 0.021), meropenem (*p* = 0.002), amoxicillin/clavulanic acid (*p* = 0.017), cefepime (*p* = 0.038), and trimethoprim/sulfamethoxazole (*p* = 0.0056).

### 2.5. Multidrug-Resistant Bacteria Involvement

Of the total bacteria identified in both patient groups, 51 (21.16%) of 241 in the CCH group and 15 (7.73%) of 194 in the HLH group were classified as MDR bacteria (*p* < 0.0001). Table 6 provides a detailed breakdown of the types and categories of these bacteria. In the HLH group, the vast majority of MDR bacteria (14 of 15) were of the ESBL (extended-spectrum beta-lactamase) or MLS (macrolide–lincosamide–streptogramin) types.

Conversely, in the CCH group, the bacterial spectrum was more diverse, including resistance types such as ESBL, CPE (carbapenemase-producing *Enterobacterales*), VRE (vancomycin-resistant *Enterococci*), and HLAR (high-level aminoglycoside resistance). Due to the distinct number of MDR bacteria observed between the two groups, a bivariate comparative analysis could not identify significant differences in the overall bacterial characteristics. However, in both groups, the majority of the MDR bacteria were isolated from patients with biliary stents. This association was statistically significant in the CCH group (*p* < 0.0001).

A particularity of our study was the ability to observe some of the included patients over multiple episodes of AC, thus allowing for the dynamic evaluation of their bile culture outcomes. Therefore, during the study period, in the CCH study group, 22 of the 32 patients with multiple episodes of AC had at least one recurrence of the same bacterial species previously isolated. Of these, nine were represented by *Escherichia coli* (of which eight developed resistances to at least one class of antibiotic) and ten by *Pseudomonas* spp. (of which four developed resistances to at least one antibiotic class of interest). In the case of the HLH group, the only bacterium identified over two episodes was *E. faecium*, which developed resistance to imipenem.

### 2.6. Antimicrobial Treatment

The most common antibiotics used empirically in the CCH group (detailed in Table 7) were the ceftriaxone–metronidazole association (in 121 episodes of AC, 64.02%), followed by ceftriaxone alone (22, 11.64%), ciprofloxacin (13, 6.87%), meropenem (9, 4.76%) and piperacillin/tazobactam (5, 2.89%). Within the HLH group, piperacillin/tazobactam was the most commonly used initial antibiotic therapy (30, 44.77%), followed by amoxicillin/clavulanic acid (9, 13.43%), the combination ceftriaxone/amoxicillin/clavulanic acid, trimethoprim/sulfamethoxazole, and the piperacillin/tazobactam–amikacin combination (each used in 3 AC episodes or 4.47%). Moreover, the initial antibiotic regimen was changed in 38 (20.10%) cases of AC for the CCH group and 7 cases (10.44%) for the HLH group (*p* = 0.0925).

During hospitalization, the initial antibiotic regimen was changed for 38 (20.10%) of the AC cases in the CCH group and for 7 (10.44%) cases in the HLH group (*p* = 0.0925). A total of 26 of the 38 (68.42%–CCH) and 3 of the 7 (42.85%–HLH) antibiotic changes were made for patients who had a biliary stent already in place. The most frequently used second line antibiotic in the CCH group was meropenem (n = 12), imipenem/cilastatin (n = 10), and the meropenem–metronidazole combination (n = 3), while in the HLH group, the most commonly used second-line antibiotics were piperacillin/tazobactam (n = 2), piperacillin/tazobactam–meropenem (n = 1), and the piperacillin/tazobactam–linezolid combination (n = 1).

### 2.7. Risk Factors Associated with MDR Bacteria Infections

After performing a logistic regression analysis (Table 8) for each study group (as previously described in Section 2.4), we identified several RFs associated with an MDR bacterial infection. In the CCH group, the identified RFs were, in order of strength of association, resistance to at least two fluoroquinolones, with an odds ratio (OR) of 6.25 (confidence interval (CI) between 2.817 and 13.869), *p* < 0.001, resistance to at least two III or IV generation cephalosporins, OR = 6.009 (CI between 3.084 and 11.708), *p* < 0.001, resistance to carbapenems with an OR of 4.45 (CI between 1.803 and 10.982), *p* = 0.001, biliary stent presence, OR of 2.847 (CI between 1.377 and 5.888), *p* = 0.005, and a positive blood culture, OR = 0.646 (CI between 0.212 and 1.967), *p* = 0.042.

In the HLH study group, the only RF proven to be associated with the presence of MDR bacterial infection were resistance to at least two fluoroquinolones, OR = 19.444 (CI between 4.752 and 79.566), *p* < 0.001 resistance to piperacillin/tazobactam, OR = 3.508 (CI between 1.136 and 10.831), *p* = 0.022, resistance to amoxicillin/clavulanic acid, OR = 3.508 (CI between 1.136 and 10.831), *p* = 0.029, and resistance to at least two third or fourth generation cephalosporins, OR = 3.107 (CI between 0.013 and 9.533), *p* = 0.047.

## 3. Discussion

Our study analyzed the microbiological profile and the AR patterns from routine bile aspirates in the AC patients based on data from two reference centers in Romania and France. The research was based on two main hypotheses. First, the AR rates would be higher in a country like Romania, where antibiotic consumption is significantly higher and current health policies are not yet sufficiently implemented. The second assumption was that resistance rates would be predominantly higher in patients with a history of endoscopic sphincterotomy, considering the altered anatomy and the potential presence of a biliary stent, both of which can alter the local biliary microbiota.

A number of programs aimed at limiting antibiotic consumption, both at home and in hospitals, have been adopted in both countries in recent years. Since 2006, Romania has prohibited the purchase of antibiotics without prescription, and since 2023, has launched the National Strategy for Limiting Antimicrobial Resistance and Preventing Healthcare Associated Infections, which includes such measures as limiting the number of emergency antibiotics available without prescription, introduction of rapid testing for common bacterial infections, including acute pharyngeal streptococcal infection and mandatory daily reporting, and centralization of antibiotic sales by pharmacists [23,24,25]. Similarly, France has introduced a series of measures to reduce the national antibiotic consumption by 25% by 2025, such as prescribing first-line medications and avoiding broad-spectrum antibiotics, and reimbursement of antibiotics only in the presence of a positive rapid test [26,27,28].

### 3.1. Risk Factors Associate with MDR Bacteria Infections

Our findings suggest that *E. coli*, *Pseudomonas* spp., *Klebsiella* spp., and *Enterococcus* were the most common microbial agents in Romania. By contrast, the most prevalent pathogens in France were *Enterococcus*, *E. coli*, and *Streptococcus*.

Similarly, a German investigation also reported Gram-negative bacteria as the most common agents in AC, in over 60% of bile samples. In their assessment, blood culture positivity was 32% [29]. Another investigation, which involved 208 individuals, revealed that ~85% of bile samples were positive for at least one microbial agent, with *E. faecalis* (~33%), *E. faecium* (~29%), *E. coli* (~23%), and *Klebsiella* spp. (~10%) as the most common. A polymicrobial cultures analysis revealed the prevalence of Gram-positive bacteria at ~73%, Gram-negative at ~53%, and fungi at ~24% (*Candida albicans* ~20%) [30].

Another US-based study on 615 subjects suspected of having AC also demonstrated a positivity of over 90% for bile aspirate samples, of which over 80% were polymicrobial, and for approximately 40% of blood cultures. Interestingly, Gram-positive agents were more commonly detected compared to our assessment, with *Enterococcus* spp. affecting nearly 70% and *Streptococcus viridans* over 35% of cases. Prior biliary endoscopic sphincterotomy was associated with a 4.5 times higher rate of bile culture positivity, as well as elevated rates of *Enterococcus* spp. (OR: 6.0), *Pseudomonas aeruginosa* (OR: 4.8), *Enterobacter* spp. (OR: 2.9), VRE (OR: 2.5), and *Klebsiella* spp. (OR: 2.1), respectively. Similarly, a history of stent placement was linked to elevated rates of *Enterococcus* spp. (OR: 9.9), *Pseudomonas aeruginosa* (OR: 5.2), *Enterobacter* spp. (OR: 3.9), VRE (OR: 2.8), and *Klebsiella* spp. (OR: 2.7) [9].

Other studies conducted on AC cases from Romania have reported similar results, but also contrasting findings. In an investigation conducted on 488 AC patients, blood culture positivity was less than 40%, whereas bile cultures were positive mainly in patients with a history of gallbladder removal (81.4%) [3]. In another Romanian research, which recruited 262 AC patients, bile cultures were positive for one microbial strain in 46%, two strains in 24%, and three strains in 3%, respectively. Gram-negative bacteria were also notably prevalent, the authors reporting the presence of *E. coli* predominantly in patients with benign vs. malignant biliary obstruction (56.1% vs. 37.6%). *Klebsiella*, *Pseudomonas*, and *Citrobacter* were identified in 24–29%, 10–14%, and 4.7–7% of cases, whereas *Enterococcus* was depicted in 18.7–24.7% of the AC subjects [1].

Similarly, *E. coli* and *Klebsiella* spp. strains were also the most detected Gram-negative agents in a cohort study conducted in Iran. In terms of Gram-positive strains, *Staphylococcus aureus* was the most commonly identified germ, as opposed to *Enterococcus* in our assessment [31]. Interestingly, an investigation performed in South Korea identified monomicrobial infections in 98.1% of the AC cases, revealing an increasing trend in the detection of Gram-negative agents, particularly *Enterobacterales*, and a decreasing trend of Gram-positive agents, namely *Enterococci*, over time [32].

Another study conducted in China on 277 patients with biliary tract infections reported that Gram-negative bacteria represented over 75% of the microorganisms identified in bile samples, with *E. coli* and *K. pneumoniae* comprising 26% and 15.7% of the microbial isolates, respectively. In addition, as opposed to our study, they identified a nearly 5% prevalence of *Pseudomonas* and *Enterobacter* each, whereas in our assessment *Pseudomonas* was identified in nearly 30% of the samples in Romania and less than 3% in France, and *Enterobacter* was reported in approximately 3% of the bile samples collected in the Romanian hospital and 8% of those collected in the French one. Related to Gram-positive agents, *Enterococcus* was the most frequently detected pathogen in both the Chinese investigation (12%) and ours, followed distantly by *Staphylococcus* spp. and *Streptococcus* spp. (approximately 4% and 2%, respectively) [14].

### 3.2. Antimicrobial Resistance Rates

In terms of AMR, resistance to multiple antibiotic classes was most commonly detected in the AC subjects from Romania who displayed resistance to multiple antibiotics (ceftriaxone, cefazoline, ciprofloxacin, or levofloxacin), whereas microbial strains in France were more likely resistant to amoxicillin/clavulanate and piperacillin/tazobactam. Resistance to at least two fluoroquinolones, III/IV generation cephalosporins, and carbapenems, as well as stent placement and blood cultures positivity, emerged as RFs for MDR AC, according to the data collected from the Romanian hospital. Similarly, the detection of MDR AC in the French hospital was predicted by the presence of resistance to fluoroquinolones, III/IV generation cephalosporins, amoxicillin/clavulanate, and piperacillin/tazobactam.

The same previously mentioned German study showed similar AR rates to the French reference center we examined, but significantly lower than in Romania, where bacterial strains seem resistant to multiple broad-spectrum antibiotics. The AR of *Enterobacterales* to ampicillin/sulbactam was over 50%, 34% to piperacillin/tazobactam, and between 20% and 27% to ciprofloxacin and cefotaxime. In addition, male sex, a high comorbidity burden, and percutaneous procedure emerged as RFs for the detection of MDR gram-negative bacteria [29].

In the Romanian cohort, more than 44% of the patients who had undergone cholecystectomy displayed ESBL-bacteria in the bile, 16.3% carbapenem-resistant *Enterobacteriaceae*, and 14.0% VRE, with MDR bacteria detected in nearly 75% of these subjects as opposed to 31.5% in the AC patients with intact gallbladders. In terms of microbial strains, *E. coli* and *Klebsiella* spp. were represented in percentages similar to our Romanian hospital; however, we detected a higher incidence of *Pseudomonas* (28% versus <7%) and a lower incidence of *Enterococcus* (15% versus 23.6%). AMR displayed a similar pattern, with 20–30% of the cases resistant to ampicillin/sulbactam, piperacillin/tazobactam, II and IV generation cephalosporins, and 15–20% resistant to fluoroquinolones and carbapenems. In another study, AC patients with biliary obstruction of malignant etiology displayed significantly higher rates of resistance to imipenem, ceftazidime, cefepime, and meropenem versus those with a benign etiology [2].

An assessment conducted in Turkey pointed out that *E. coli* (28.2%) and *Pseudomonas* (17.3%) are the most commonly identified pathogens in AC; however, they also revealed a high prevalence of *Stenotrophomonas maltophilia* at 15.2%. AMR patterns were commonly detected in both Gram-negative (*P. aeruginosa* and *E. coli*) as well as Gram-positive (*E. faecium*), with many strains showing below or around 50% susceptibility to ampicillin/sulbactam, cefotaxime, ampicillin, fluoroquinolones, or gentamicin [13].

In our study, bile cultures showed a positivity rate of 52%. This proportion is slightly lower than other reports in the literature, with rates that range between 60% and 87% [28,29,30]. There are some possible factors associated with this phenomenon. (1) Prior antibiotic treatment, which is negatively correlated with bile culture positivity, as it reduces bacterial load and may lead to false-negative cultures. (2) Technical limitation, such as insufficient bile aspirate, or delays in transport and processing can ultimately compromise culture viability—particularly for anaerobic or fastidious organisms. While sample processing protocols were strictly followed, these factors cannot be neglected. (3) Host and disease-specific factors, including microbial concentrations which may be below detection threshold for standard cultures in immunocompetent patients, subclinical infection, or, in mild stages of AC, may not yet have reached the threshold. Similarly, malignant and benign obstructions show different colonization patterns and culture yields, which further influence the culture results. (4) Lack of advanced diagnostic methods, such as 16S rRNA sequencing or polymerase chain reaction, which have superior sensitivity compared to standard cultures. However, these technologies are not routinely available or require further clinical validation; hence, they are primarily used in research facilities and not clinical hospitals [1,13,33,34,35,36,37,38].

Our study has several strengths and limitations. Firstly, to our knowledge, it is the first comparative study between two reference gastroenterology centers in Eastern Europe (Romania) and Western Europe (France), both with significant expertise in the management of AC. Moreover, our study was prospective in design, and thus causality can be established based on our findings. However, it only includes two centers and thus, a limited number of patients. Moreover, since it was conducted in university reference centers, our assessment might also be subject to selection bias and does not generally reflect the reality if superposed to other local or regional hospitals. However, our investigation raises awareness regarding the elevated AMR rates in AC, particularly in Romania, and can serve as documentation for policymakers in the healthcare field to establish regional or national guidelines to guide the selection of empiric antibiotic regimens in AC.

## 4. Materials and Methods

### 4.1. Study Design and Ethical Considerations

We conducted a prospective, multicenter, observational cohort study in which we included patients diagnosed with AC with a positive bile sample culture and subsequent antibiogram from two tertiary healthcare facilities from Romania and France: Gastroenterology Department of Colentina Clinical Hospital (CCH) in Bucharest, Romania and Gastroenterology, Hepatology and Digestive Oncology Department of Haut-Lévêque Hospital (HLH) in Bordeaux, France between April 2022 and October 2023. The study protocol conformed to the ethical guidelines of the 1975 Declaration of Helsinki and was approved by the internal review board of both medical centers. We obtained informed consent from all the included patients before data collection, thus ensuring strict confidentiality and privacy of their information. This study was approved by the Hospital’s Ethics Committee under registration number 10/12.09.2022.

### 4.2. Patient Selection and Bile Samples Collection

All of the patients included in this study had a positive diagnosis of AC according to the latest 2018 Tokyo Guidelines (TG18) criteria, and a positive bile culture sample obtained during an ERCP or an ultrasonographic-guided percutaneous transhepatic cholangiography (PTC) [39]. The exclusion criteria included patients who did not sign the informed consent, patients aged <18 years, and patients who were enrolled in another randomized clinical trial. Some eligible patients experienced multiple episodes of AC during the inclusion period; therefore, all episodes were considered. Patients were also categorized according to their endoscopic history.

We used the Olympus 190 series duodenoscopes (Olympus Corp., Tokyo, Japan), which were disinfected according to the manufacturer’s instructions, and according to the latest European Society of Gastrointestinal Endoscopy (ESGE)/European Society of Gastroenterology Nurses and Associates (ESGENA) position statement on the prevention of multidrug-resistant infections from contaminated duodenoscopes and to the latest guidelines of the French Ministry of Health and the French Society of Gastrointestinal Endoscopy [40,41,42,43]. Contamination was systematically excluded through standardized weekly microbiological tests according to local epidemiological health standards and protocols. Bile samples were collected after selective cannulation of the CBD with a standard single-use 4.4F TRUEtome^TM^ sphincterotome and a guide-wire (Boston Scientific, Marlborough, MA, USA). Approximately 5–10 mL of bile was aspirated before the injection of the contrast agent, and then placed in sterile containers suited for both anaerobic and aerobic bacterial cultures. Samples were transferred to the microbiology department and then cultured under aerobic and anaerobic conditions at 37 °C on 5% Columbia sheep blood agar (Becton Dickinson GmbH, Heidelberg, Germany), and Sabouraud dextrose agar (Thermo Fisher Scientific™, Waltham, MA, USA) with a first reading after 24 h, and the second reading at 48 h. Aerobic microorganisms were routinely identified and tested for antibiotic susceptibility by determining the minimum inhibitory concentration using the VITEK 2**^®^** System (bioMérieux, Lyon, France), while the identified fungi were tested using either the VITEK 2 YST system (bioMérieux, France) or the BD Phoenix™ yeast ID system (Becton, Dickinson and Company, Franklin Lakes, NJ, USA), all according to the latest European Committee on Antimicrobial Susceptibility Testing (EUCAST—https://www.eucast.org, accessed on 19 January 2025) guidelines available at the time.

### 4.3. Data Acquisition and Study Variables

Eligible patients were asked to sign the informed consent after each endoscopic procedure and only after being thoroughly informed about the purpose and implications of this study. Data was collected using a standardized data collection form from the electronic hospital database, patients’ charts, discharge reports, as well as imaging and laboratory files, during the hospital stay or after their discharge. The following variables were included: demographics (age, sex), clinical history (previous hospitalizations, comorbidities, medications, previous endoscopic interventions, history of cholecystectomy), presenting symptoms, laboratory data (blood count, C-reactive protein (CRP), total and direct bilirubin, aspartate aminotransferase (AST) and alanine aminotransferase (ALT)—all reported to laboratory reference values), ERCP timing, hospital stay duration, Tokyo severity score, the etiology of AC, admission rate to the intensive care unit (ICU), empiric and guided antibiotic regimes, bile culture results, and the subsequent AR pattern and MDR status assessment. Complications were also noted and further categorized according to the latest Adverse Events for Gastrointestinal Endoscopy (AGREE) classification [44].

### 4.4. Statistical Analysis

Statistical analyses were performed using the SPSS software, version 29.0 (IBM, Endicott, New York, NY, USA) and GraphPad Prism v9.2.0, graphical software (GraphPad, San Diego, CA, USA). Continuous variables were presented as a mean (with standard deviation), while categorical variables were reported as the number of subjects (n) and the percentage (%). Descriptive statistics showed the main demographic and clinical characteristics. The microbial species distribution and the AR patterns between the groups and sub-groups were compared using the chi-squared test or Fisher’s exact test, with a *p*-value < 0.05 considered to be statistically significant. Continuous variables’ normality was assessed using the D’Agostino–Pearson test, which indicated a non-normal distribution. Hence, comparisons were performed using the nonparametric Mann–Whitney U-test. In the end, we performed a logistic regression analysis in order to assess which of the RFs postulated in the literature, such as age > 75 years old, male sex, malignant stenosis, or stented CBD (independent variables), are associated with an MDR bacterial infection (dependent variable). All results were cross-verified to guarantee data accuracy.

## 5. Conclusions

In conclusion, AC remains a potentially fatal infectious disease without prompt antibiotic and interventional treatment. Our study demonstrates both significant variability in the microbiological spectrum and in the AR patterns between geographical regions that differ culturally, and in terms of health regulations. Furthermore, it highlights the influence of previous endoscopic interventions and stent placement on AR patterns and MDR bacteria incidence, thus pointing to the importance of bile sample cultures in guiding antibiotic treatment. While this study alone may not be sufficient for data extrapolation due to the small number of patients involved and the regional character of the study groups, it can serve as a basis for future extended multicenter epidemiological studies whose “real-life” results may modify international recommendations in this area.

## Figures and Tables

**Figure 1 antibiotics-14-00679-f001:**
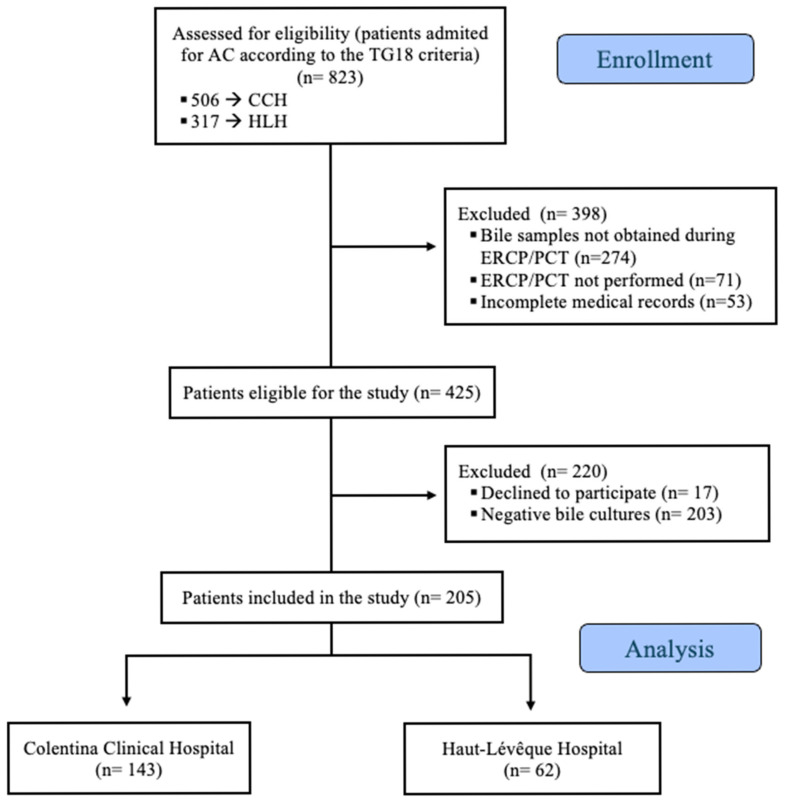
Patient selection flowchart for this study.

**Figure 2 antibiotics-14-00679-f002:**
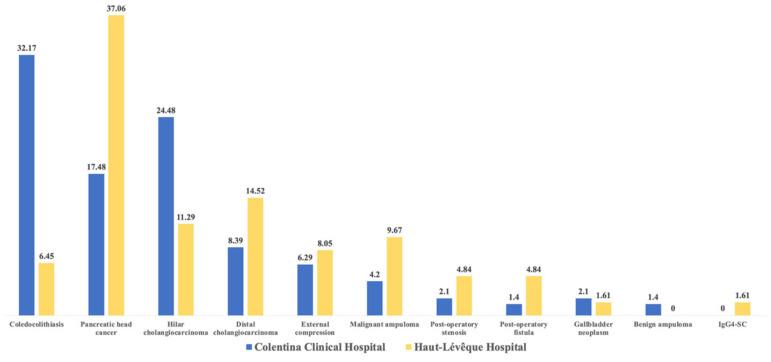
The main etiologies of acute cholangitis (%). IgG4-SC/IgG4-related sclerosing cholangitis.

**Figure 3 antibiotics-14-00679-f003:**
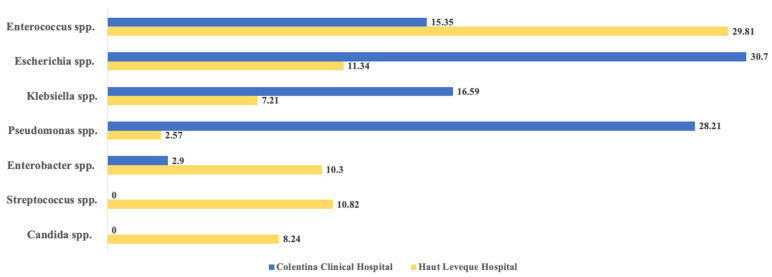
Main causative agent illustrated as percentages for each medical unit (%).

**Figure 4 antibiotics-14-00679-f004:**
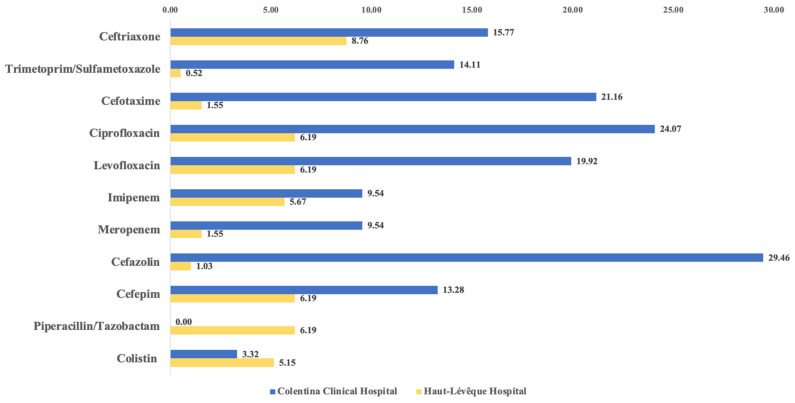
Resistance rates to the most common antibiotics.

**Figure 5 antibiotics-14-00679-f005:**
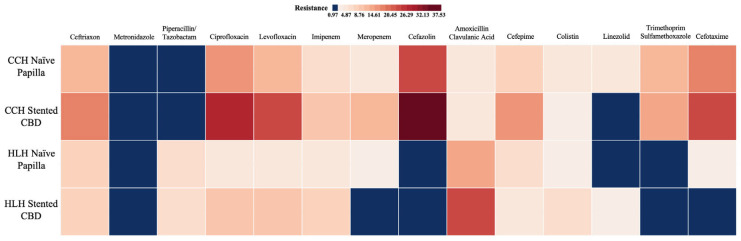
Heatmap of antibiotic resistance according to endoscopic history and clinical center.

**Table 1 antibiotics-14-00679-t001:** Detailed clinical and paraclinical characteristics of the patients included in this study and the related episodes of AC.

Variables	Colentina Clinical Hp.			Haut-Lévêque Hp.			Comparison Between Totals (*p*-Value)
	Total (n = 143)	Naive Papilla (n = 66)	Stented CBD (n = 77)	Total (n = 62)	Naive Papilla (n = 41)	Stented CBD (n = 21)	
Male sex (%)	83 (58.04)	38 (57.57)	45 (58.44)	40 (64.51)	23 (56.09)	17 (80.95)	0.5355
Mean age (SD)	68.49 (10.40)	67.71 (10.49)	69.15 (10.34)	70.34 (9.02)	70.24 (9.52)	70.58 (8.16)	0.7243
Malignant Stenosis (%)	83 (58.04%)	7 (10.60)	60 (77.92)	47 (75.80)	31 (75.60)	16 (76.19)	0.0179
Cholecystectomy (%)	41	14	27	27	18	9	0.0521
Episodes of AC							
Variables	Colentina Clinical Hp.			Haut-Lévêque Hp.			
	Total n = 189	Naive Papilla n = 86	Stent N = 103	Total N = 67	Naive Papilla N = 42	Stent N = 25	
Hospitalization days. (mean)	5.64	5.59	5.67	3.35	3.38	3.32	<0.0001
Complications. yes	57 (30.15%)	26 (30.23%)	31 (30.09%)	13 (19.40%)	7 (16.66%)	6 (24%)	0.1107
	Tokyo Severity score						
I	140 (74.07%)	68 (79.06%)	72 (69.90%)	45 (67.16%)	28 (66.66%)	17 (68%)	0.3406
II	33 (17.46%)	14 (16.27%)	19 (18.44%)	15 (22.38%)	10 (23.80%)	5 (20%)	0.3686
III	16 (8.46%)	4 (4.65%)	12 (11.65%)	7 (10.44%)	4 (9.52%)	3 (12%)	0.6237
Need for reintervention in the same hospitalization	22 (11.64%)	9 (10.46%)	13 (12.62%)	2 (2.98%)	1 (2.38%)	1 (4%)	0.0484
ICU admission	6 (3.17%)	1 (1.16%)	5 (4.85%)	4 (5.97%)	2 (4.76%)	2 (8%)	0.2941
Need for readmission < 72 h after initial discharge	4 (2.11%)	1 (1.16%)	3 (2.91%)	0	0	0	0.5755
Initial neg culture in previous episode	44	2	42	No data			Not possible
Positive/Negative blood cultures	19/18	10/7	9/11	22/9	14/4	8/5	
WBC (mean value) (/mm^3^)	12,388	12,100	12,630	11,861	11,519	12,437	>0.9999
TB (mean value) (mg/dL)	8.18	9.39	7.17	7.07	6.92	7.31	>0.9999
CRP (mean value)	91.94	74.51	106.51	85.87	83.26	90.24	0.8821

Abbreviations: WBC—white blood cells; CRP—C-reactive protein; TB—total bilirubin; ICU—intensive care unit; SD—standard deviation.

**Table 2 antibiotics-14-00679-t002:** Number of bacteria identified in each bile culture.

	CCH			HLH			Comparison Between Totals (*p* Value)
Variables	Total (n = 190)	Naive Papilla (n = 78)	Stented CBD (n = 112)	Total (n = 67)	Naive Papilla (n = 42)	Stented CBD (n = 25)	
1 bact	142	63	79	18	8	10	<0.0001
2 bact	45	14	31	13	10	3	0.5026
3 bact	3	1	2	12	5	7	<0.0001
≥4 bact	0	0	0	24	19	5	<0.0001

**Table 3 antibiotics-14-00679-t003:** The main causative agents isolated in bile cultures.

Causative Agents	Colentina Clinical Hp.			Haut-Lévêque Hp.			Comparison Between Totals (*p* Value)
	Total (n = 241)	Naive Papilla (n = 94)	Stented CBD (n = 147)	Total (n = 194)	Naive Papilla (n = 135)	Stented CBD (n = 59)	
*Enterococcus* spp.	37 (15.35%)	14 (14.89%)	23 (15.64%)	58 (29.81%)	40 (29.62%)	18 (30.50%)	0.0003
*Escherichia* spp.	74 (30.70%)	24 (25.53%)	50 (34.01%)	22 (11.34%)	17 (12.59%)	5 (8.47%)	<0.0001
*Klebsiella* spp.	40 (16.59%)	11 (11.70%)	29 (19.72%)	14 (7.21%)	11 (8.14%)	3 (5.08%)	0.0033
*Enterobacter* spp.	7 (2.90%)	5 (5.32%)	2 (1.36%)	20 (10.30%)	12 (8.88%)	8 (13.55%)	0.0022
*Streptococcus* spp.	0 (0%)	0 (0.0%)	0 (0.0%)	21 (10.82%)	12 (8.88%)	9 (15.25%)	<0.0001
*Candida* spp.	0 (0%)	0 (0.0%)	0 (0.0%)	16 (8.54%)	12 (8.88%)	4 (6.77%)	<0.0001
*Pseudomonas* spp.	68 (28.21%)	33 (35.10%)	35 (23.80%)	5 (2.57%)	4 (2.96%)	1 (1.69%)	<0.0001

**Table 4 antibiotics-14-00679-t004:** Resistance rates to the most common antibiotics.

	CCH (n = 241)	HLH (n = 194)	*p*-Value
Ceftriaxone	38 (15.77%)	17 (8.76%)	0.030
Piperacillin/Tazobactam	0 (0.00%)	12 (6.19%)	<0.0001
Ciprofloxacin	58 (24.07%)	12 (6.19%)	<0.0001
Levofloxacin	48 (19.92%)	12 (6.19%)	<0.0001
Meropenem	23 (9.54%)	3 (1.55%)	0.0003
Cefazolin	71 (29.46%)	2 (1.03%)	<0.0001
Amoxicillin/Clavulanic Acid	11 (4.56%)	42 (21.65%)	<0.0001
Colistin	8 (3.32%)	10 (5.15%)	0.346
Linezolid	6 (2.49%)	3 (1.55%)	0.737
Trimethoprim/Sulfamethoxazole	34 (14.11%)	1 (0.52%)	<0.0001
Cefotaxime	51 (21.16%)	3 (1.55%)	<0.0001

**Table 5 antibiotics-14-00679-t005:** Resistance rates to the most common antibiotics based on prior endoscopic history in the Colentina Clinical Hospital, the Haut-Lévêque Hospital, and compared between the two centers.

	Colentina Clinical Hospital (n = 241)			Haut-Lévêque Hospital (n = 194)			*p*-Value N-CCH vs. N-HLH	*p*-Value S-CCH vs. S-HLH
	Naïve Papilla (n = 94)	Stented CBD (n = 147)	*p*-Value N-CCH vs. S-CCH	Naïve Papilla (n = 136)	Stented CBD (n = 58)	*p*-Value N-HLH vs. S-HLH		
Ceftriaxone	12 (12.77%)	26 (17.69%)	0.366	12 (8.82%)	5 (8.62%)	>0.9999	0.383	0.130
Piperacillin/Tazobactam	0 (0.0%)	0 (0.0%)	>0.9999	8 (5.88%)	4 (6.9%)	0.753	0.022	0.0059
Ciprofloxacin	16 (17.02%)	42 (28.57%)	0.045	6 (4.41%)	6 (10.34%)	0.188	0.002	0.0056
Levofloxacin	11 (11.7%)	37 (25.17%)	0.012	6 (4.41%)	6 (10.34%)	0.188	0.043	0.021
Imipenem	7 (7.45%)	16 (10.88%)	0.501	6 (4.41%)	5 (8.62%)	0.309	0.389	0.799
Meropenem	5 (5.32%)	18 (12.24%)	0.113	3 (2.21%)	0 (0.0%)	0.555	0.276	0.002
Cefazolin	35 (37.23%)	36 (24.49%)	0.042	2 (1.47%)	0 (0.0%)	>0.9999	>0.9999	>0.9999
Amoxicillin/Clavulanic Acid	4 (4.26%)	7 (4.76%)	>0.9999	33 (24.26%)	9 (15.52%)	0.189	>0.9999	0.017
Cefepime	8 (8.51%)	24 (16.33%)	0.118	9 (6.62%)	3 (5.17%)	>0.9999	0.615	0.038
Colistin	5 (5.32%)	3 (2.04%)	0.267	8 (5.88%)	2 (3.45%)	0.726	>0.9999	0.622
Linezolid	5 (5.32%)	1 (0.68%)	0.034	3 (2.21%)	0 (0.0%)	0.555	0.276	>0.9999
Trimethoprim/Sulfamethoxazole	12 (12.77%)	22 (14.97%)	0.706	0 (0.0%)	1 (1.72%)	0.298	>0.9999	0.0056
Cefotaxime	23 (24.47%)	28 (19.05%)	0.335	3 (2.21%)	0 (0.0%)	0.555	>0.9999	>0.9999

Abbreviations: N-CCH—Naïve papilla Colentina Hospital; N-HLH—Naïve papilla Haut-Lévêque Hospital; S-CCH—Stented CBD Colentina Hospital; S-HLH—Stented CBD Haut-Lévêque Hospital, CBD—common bile duct. Significance was calculated using the chi-squared test and Fisher’s exact test, respectively, to evaluate resistance rates to different antibiotics between patients with a naïve papilla and with a stented common bile duct at the Colentina Clinical Hospital, Romania, and at the Haut-Lévêque Hospital, France, as well as between patients with a naïve papilla in Romania vs. France, as well as patients with a stented common bile duct in Romania vs. France.

**Table 6 antibiotics-14-00679-t006:** The main types of MDR bacteria.

MDR Agents	Colentina Clinical Hp.			Haut-Lévêque Hp.		
	Total 51	Naive Papilla 11	Stent 40	Total 15	Naive Papilla 7	Stent 8
Extended-spectrum beta-lactamase (ESBL)	28	4	24	6	4	2
Carbapenemase-producing *Enterobacterales* (CPE)	7	1	6	0	0	0
Carbapenemase-producing *Enterobacterales* New Delhi metallo-β-lactamase (CPE-NDM)	2	0	2	0	0	0
Vancomycin-resistant *Enterococci* (VRE)	5	3	2	0	0	0
Macrolide–lincosamide–streptogramin B antibiotics resistant (MLS)	0	0	0	8	2	6
High level aminoglycoside resistant *Enterococci* (HLAR)	7	2	5	0	0	0
Others *	2	1	1	1	1	0

* Others—Citrobacter freundii MDR; Acinetobacter baumannii XDR.

**Table 7 antibiotics-14-00679-t007:** The main antibiotic used as a first-line treatment.

Antibiotics Used as First-line Treatment	Colentina Clinical Hp.			Haut-Lévêque Hp.		
	Total	Naïve Papilla	Stented CBD	Total	Naive Papilla	Stented CBD
Ceftriaxone + Metronidazole	121	43	78	2	0	2
Ceftriaxone	22	9	13	0	0	0
Ciprofloxacin	13	9	4	0	0	0
Meropenem	9	2	7	2	2	0
Amoxicillin/Clavulanic Acid	5	4	1	9	5	4
Piperacillin/Tazobactam	0	0	0	30	19	11

**Table 8 antibiotics-14-00679-t008:** Risk factors involved in MDR bacteria involvement.

Variables	Odds Ratio (Lower CI–Upper CI)	*p*-Value	Odds Ratio (Lower CI–Upper CI)	*p*-Value
	CCH		HLH	
Age > 75 years old	1.232 (0.642–2.356)	0.531	1.057 (0.34–3.292)	0.923
Male sex	1.137 (0.606–2.136)	0.689	0.643 (0.194–2.132)	0.471
Polymicrobial bile cultures	1.786 (0.954–3.343)	0.700	1.356 (0.167–11.01)	0.776
Positive blood culture	2.646 (0.212–1.967)	0.042	0.231 (0.029–1.814)	0.163
Complications	0.678 (0.331–1.388)	0.288	3.148 (0.98–10.116)	0.054
Malignant stenosis	1.664 (0.815–3.397)	0.162	0.537 (0.139–2.071)	0.367
Resistance to ≥2 Fluoroquinolones	6.25 (2.817–13.869)	<0.001	19.444 (4.752–79.566)	<0.001
Resistance to Carbapenems	4.45 (1.803–10.982)	0.001	Not significant	
Resistance to ≥2 III/IV Cephalosporins	6.009 (3.084–11.708)	<0.001	3.107 (0.013–9.533)	0.047
Resistance to Piperacillin/Tazobactam	1.985 (0.894–4.405)	0.092	3.99 (1.225–12.994)	0.022
Resistance to Amoxicillin/Clavulanic Acid	2.213 (0.622–7.876)	0.22	3.508 (1.136–10.831)	0.029
Hospital stay	0.969 (0.881–1.065)	0.51	1.24 (0.939–1.636)	0.129
ICU admission	0.61 (0.072–5.185)	0.651	Not significant	
Need for reintervention	0.87 (0.311–2.432)	0.79	Not significant	
Stent placement	2.847 (1.377–5.888)	0.005	2.462 (0.823–7.366)	0.107
Tokyo score of III	0.861 (0.277–2.682)	0.059	Not significant	

Significance was obtained using multivariate logistic regression to evaluate the risk factors for the development of multidrug-resistant bacteria (MDR) in two gastroenterology reference centers, the Colentina Clinical Hospital (CCH) in Romania, and the Haut-Lévêque Hospital (HLH) in France. *p*-values equal to or less than 0.05 were considered statistically significant and reported using the green color.

## Data Availability

Data is contained within the article.

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
