# Peer review of "Microbial Profile and Antibiotic Resistance Patterns in Bile Aspirates from Patients with Acute Cholangitis: A Multicenter International Study"

_antibiotics, 2025, doi:10.3390/antibiotics14070679_

Round 1
Reviewer 1 Report
Comments and Suggestions for Authors
I extend my appreciation to the authors for their study, which suggests significant differences between countries in the profile of microorganisms growing in bile cultures of patients with acute cholangitis and in the antibiotic resistance patterns of these microorganisms. This work is a valuable contribution to the field of microbiology and infectious diseases.
Could the authors provide more details on whether patients' current antibiotic treatment was a criterion in patient selection? This information is crucial for understanding the study's methodology.
It would be beneficial to understand how the potential effect of antibiotic treatment, initiated without obtaining a bile sample for culture via ERCP or PTC, was evaluated on the results. This information is key to interpreting the study's findings.
Were AC patients with no growth in bile culture evaluated separately?
Of the 425 patients deemed eligible for the study, 203 were excluded because no growth was observed in bile cultures. How do you interpret the fact that approximately 48% of patients had no growth in their bile cultures?
Why were blood cultures taken from only 68 of patients when AC is a life-threatening infection? Are there differences in blood culture collection policies among hospitals?
Could you provide details on the antibiotics used in patients diagnosed with AC?
In your cohort, which reported a significant morbidity rate after ERCP, how do you evaluate the average length of stay of 5.64 and 3.38 days, given that there was also substantial antimicrobial resistance in the growing microorganisms? How many days did you typically continue the initiated antibiotic treatment, and did you administer parenteral antibiotic treatment after discharge in some patients?
Author Response
Dear Editor-in-Chief, Guest Editors, and Peer-Reviewers,
We are very thankful to you for the pertinent notes; we have carefully read the comments and have revised/completed the manuscript accordingly. Our responses are given in a point-by-point manner below. All the changes to the manuscript are highlighted in yellow. We hope that, in this new form, the manuscript will be suitable for publication in Antibiotics.
I extend my appreciation to the authors for their study, which suggests significant differences between countries in the profile of microorganisms growing in bile cultures of patients with acute cholangitis and in the antibiotic resistance patterns of these microorganisms. This work is a valuable contribution to the field of microbiology and infectious diseases.
Response: Thank you for your positive feedback regarding our manuscript and for considering that the results of our study are worth publishing in Antibiotics.
Could the authors provide more details on whether patients' current antibiotic treatment was a criterion in patient selection? This information is crucial for understanding the study's methodology.
Response: Thank you for your comment. We confirm that ongoing antibiotic treatment at the time of admission was not used as a selection criterion for patients included in our study. A proportion of the patients had been transferred from other medical facilities and had already received empiric antibiotic therapy before admission to the centers where our study was conducted. However, neither the administration of materials nor the specific type of antibiotic used before admission influenced inclusion in the study cohort. As specified in the Materials and Methods section, the only inclusion criteria were a confirmed diagnosis of acute cholangitis and the presence of a positive bile culture.
It would be beneficial to understand how the potential effect of antibiotic treatment, initiated without obtaining a bile sample for culture via ERCP or PTC, was evaluated on the results. This information is key to interpreting the study's findings.
Response: Thank you for raising this valuable point. Indeed, empiric antibiotic therapy initiated prior to bile culture collection can significantly influence microbiological results by suppressing bacterial growth and potentially altering the observed antimicrobial resistance profiles. Although prior antibiotic administration was common in our patient cohort (particularly among patients transferred from other medical centers) it was not used as a selection criterion and thus was not systematically controlled or recorded for detailed subgroup analysis. We acknowledge this as an important limitation of our study and have now explicitly addressed it in the revised manuscript’s Discussion and Limitations sections. To asses the potential effect of empiric antibiotic regimes, future prospective randomised studies, with systematic documentation, subgroup analyses based on antibiotic exposure, and at least two groups of patients (with or without antibiotic exposure prior tu bile sampling) would be beneficial to further explore and clarify this aspect. However, these studies are difficult to perform, as immediate antibiotic treatment is mandatory in all patients with suspected AC.
Were AC patients with no growth in bile culture evaluated separately?
Response: Thank you for your comment. Patients with negative bile cultures were not included in our study and, therefore, were not evaluated separately. As specified in the inclusion criteria, our analysis focused exclusively on patients with a confirmed diagnosis of acute cholangitis (AC) and a positive bile culture result, in order to allow a meaningful assessment of the microbial spectrum and antimicrobial resistance patterns.
Of the 425 patients deemed eligible for the study, 203 were excluded because no growth was observed in bile cultures. How do you interpret the fact that approximately 48% of patients had no growth in their bile cultures?
Response: Thank you for your comment regarding the relatively high proportion (48%) of patients with negative bile cultures in our cohort. This is indeed an important aspect, and we have addressed it in a dedicated section of the revised manuscript. First, it is important to note that negative bile culture results are not uncommon in acute cholangitis, with rates reported in the literature ranging from 27% to 40%, even in patients with clinically and radiologically confirmed biliary infection (10.1097/MEG.0000000000002849 , . Our findings are slightly higher and can be attributed to several real-world clinical and technical factors:
- Prior antibiotic treatment: A significant number of patients in our study, particularly those transferred from other institutions, had already received empiric antibiotic therapy prior to ERCP. As documented in several studies, the duration of prior antibiotic use is negatively correlated with bile culture positivity, as antimicrobial exposure reduces bacterial load and may lead to false-negative cultures (10.3748/wjg.v18.i27.3585).
- Technical limitations: Factors such as insufficient bile volume, inadequate timing of sample collection, or delays in transport and processing can compromise culture viability, particularly for anaerobic or fastidious organisms. These organisms may require special media, enriched conditions, or extended incubation periods that are not part of standard aerobic culture protocols. While sample processing protocols were strictly followed, these factors cannot be neglected.
- Methodological constraints: Standard culture techniques may fail to detect bacteria present at low concentrations, non-culturable species, or pathogens embedded within biofilms, especially in patients with previous biliary instrumentation. Anaerobic bacteria like Clostridium perfringens or Bacteroides fragilis are frequently missed unless specifically targeted, and intermittent bacterial shedding into bile (similar to phenomena observed in blood cultures) can also lead to sampling during bacterium-free intervals (https://doi.org/10.3389/fcimb.2025.1575824).
- Host and disease-specific factors: In immunocompetent patients, subclinical infection may be contained by the host immune response, and in early or mild stages of acute cholangitis, microbial concentrations may not yet have reached the detection threshold for standard cultures. Similarly, malignant obstructions and benign conditions show different colonization patterns and culture yields.
- Lack of advanced diagnostic alternatives: Molecular methods such as 16S rRNA sequencing and multiplex PCR have demonstrated superior sensitivity and can identify bacterial DNA even in bile samples that are culture-negative using conventional techniques. However, these technologies are not routinely available or require further clinical validation, especially in differentiating contamination, dead bacteria, and clinically relevant infection. Hence, they are primarily used in research facilities and not clinical hospitals.
Why were blood cultures taken from only 68 of the patients when AC is a life-threatening infection? Are there differences in blood culture collection policies among hospitals?
Response: Thank you for your comment. Blood cultures were collected based on clinical judgment at the time of admission, and not uniformly across all patients with AC. This reflects real-world variability in practice, particularly in emergency or transfer situations where bile culture collection during ERCP may be prioritized over blood sampling, especially if antibiotics have already been initiated. Blood cultures were taken mainly from patients with severe forms of acute cholangitis or those whose clinical condition did not improve despite effective endoscopic treatment (reflected by a decrease in total bilirubin levels post-procedure). Moreover, we acknowledge that there may be institutional differences in blood culture collection protocols between the two participating centers. While both follow international guidelines, individual clinician discretion, timing of patient presentation, and logistical aspects (e.g., transfer from another unit, weekend/overnight admissions) could have contributed to the observed variation.
Could you provide details on the antibiotics used in patients diagnosed with AC?
Response: Thank you for your comment. In response, we have added a dedicated subsection providing details on the antibiotics administered to patients diagnosed with acute cholangitis. This section includes both the empiric regimens initiated upon admission and the targeted therapies adjusted according to culture results. The newly added content has been highlighted in yellow within the revised manuscript for your convenience. We appreciate your suggestion, which has helped us improve the clarity and clinical relevance of our work.
In your cohort, which reported a significant morbidity rate after ERCP, how do you evaluate the average length of stay of 5.64 and 3.38 days, given that there was also substantial antimicrobial resistance in the growing microorganisms? How many days did you typically continue the initiated antibiotic treatment, and did you administer parenteral antibiotic treatment after discharge in some patients?
Response: Thank you for your comment. We acknowledge the apparent discrepancy between the average hospital stay in the two groups. This difference can be partially explained by the organizational structure of the two medical centers involved in our study. While both hospitals function as specialized referral centers for biliopancreatic interventions, serving a network of surrounding medical units, in the Haut-Lévêque Hospital, patients were admitted for ERCP only and then transferred back to their referring hospital for continued care and monitoring, in many cases on the same day. In contrast, the Colentina Clinical Hospital (CCH) generally manages patients throughout their entire course of hospitalization, including pre-ERCP stabilization and post-procedure follow-up, which results in longer average stays.
Regarding antibiotic therapy, we ensured that all patients received antibiotics for the full duration recommended by the Tokyo Guidelines, adjusted as necessary once culture results became available. All positive bile culture results were communicated directly to the receiving facilities when patients were transferred, to allow appropriate modifications to antimicrobial regimens. In cases where patients were discharged directly, we waited for the final microbiological results before issuing prescriptions, ensuring that oral antibiotic regimens were appropriately tailored if adjustments were required. We would like to emphasize that parenteral antibiotic therapy was never prescribed for home administration. Intravenous antibiotics were continued only during hospitalization or in the context of an inter-hospital transfer, ensuring continuity of care without compromising safety.
Reviewer 2 Report
Comments and Suggestions for Authors
The present study is aimed to compare the microbial spectrum and antibiotic resistance patterns of bile pathogens in patients with acute cholangitis from two European centers, while also assessing the impact of prior endoscopic interventions on MDR infections. The findings reveal significant regional differences in pathogen profiles and resistance patterns, underscoring the value of bile cultures in guiding treatment and the need for broader multicenter studies to inform international guidelines. I recommend revision prior to publication.
- In the abstract, please clarify the number of patients and samples to avoid confusion. For example: “We included 144 patients from CCH, with 190 positive bile cultures (some patients had multiple episodes) and 241 identified bacterial strains…” This brief clarification improves the abstract’s clarity and allows it to stand alone.
- To strengthen the Introduction, consider clearly stating the specific gap your study addresses. While previous research has described the microbiological landscape and resistance patterns in acute cholangitis, most are single-center studies limited to one geographic region. What remains underexplored—and where your study makes a valuable contribution—is a direct comparison of microbial profiles and resistance trends between different healthcare systems, particularly between Eastern and Western Europe. This comparison is especially pertinent given the disparities in antibiotic usage, infection control policies, and healthcare infrastructure across regions. Furthermore, your focus on bile cultures, which are more sensitive and informative than blood cultures yet less frequently analyzed, enhances the study’s clinical relevance. By evaluating the impact of prior endoscopic interventions like stent placement on resistance profiles, your study goes beyond descriptive epidemiology and begins to explore risk factors for antimicrobial resistance. Although many of the bacterial species identified are well known, a brief mention of their clinical significance in the opening would provide helpful context and reinforce the value of their comparative analysis.
- In Figure 2, the Y-axis is currently not labelled. Please revise the figure to indicate that the Y-axis represents the percentage (%) of total cases.
- Please clarify the legends for Tables 5 and 7:
- For Table 7, specify whether the comparisons (p value) are between naïve papilla vs. stented common bile duct (CBD), between centers (CCH vs HLH), or both. Also indicate the statistical test(s) used.
- Similarly, table 7 does not specify the groups used for calculation of p- value.
- Please correct- Candida spp. are fungi and not bacteria. Additionally, Candida spp. require specific methods for accurate identification, please specify whether the VITEK 2 YST card or another validated fungal identification system was used. This clarification is important to ensure the reliability of the data reported in Table 3 and Figure 3.
- Please ensure that all scientific names of microorganisms are italicized throughout the manuscript, following standard scientific style conventions.
Author Response
Dear Editor-in-Chief, Guest Editors, and Peer-Reviewers,
We are very thankful to you for the pertinent notes; we have carefully read the comments and have revised/completed the manuscript accordingly. Our responses are given in a point-by-point manner below. All the changes to the manuscript are highlighted in yellow. We hope that, in this new form, the manuscript will be suitable for publication in Antibiotics.
The present study aims to compare the microbial spectrum and antibiotic resistance patterns of bile pathogens in patients with acute cholangitis from two European centers, while also assessing the impact of prior endoscopic interventions on MDR infections. The findings reveal significant regional differences in pathogen profiles and resistance patterns, underscoring the value of bile cultures in guiding treatment and the need for broader multicenter studies to inform international guidelines. I recommend revision prior to publication.
Response: Thank you for your positive feedback regarding our manuscript and for considering that the results of our study are worth publishing in Antibiotics.
- In the abstract, please clarify the number of patients and samples to avoid confusion. For example: “We included 144 patients from CCH, with 190 positive bile cultures (some patients had multiple episodes) and 241 identified bacterial strains…” This brief clarification improves the abstract’s clarity and allows it to stand alone.
Response: Thank you for your comment. We have corrected these errors and added the additional information in the manuscript.
- To strengthen the Introduction, consider clearly stating the specific gap your study addresses. While previous research has described the microbiological landscape and resistance patterns in acute cholangitis, most are single-center studies limited to one geographic region. What remains underexplored—and where your study makes a valuable contribution—is a direct comparison of microbial profiles and resistance trends between different healthcare systems, particularly between Eastern and Western Europe. This comparison is especially pertinent given the disparities in antibiotic usage, infection control policies, and healthcare infrastructure across regions. Furthermore, your focus on bile cultures, which are more sensitive and informative than blood cultures yet less frequently analyzed, enhances the study’s clinical relevance. By evaluating the impact of prior endoscopic interventions like stent placement on resistance profiles, your study goes beyond descriptive epidemiology and begins to explore risk factors for antimicrobial resistance. Although many of the bacterial species identified are well known, a brief mention of their clinical significance in the opening would provide helpful context and reinforce the value of their comparative analysis.
Response: Thank you for your thoughtful and constructive comment. In response, we have revised the Introduction section to explicitly highlight the specific knowledge gap our study addresses. We have also incorporated a brief rationale for the clinical relevance of analyzing bile cultures over blood cultures, noting their higher sensitivity and direct correlation with the biliary source of infection. Furthermore, we clarified the added value of evaluating prior endoscopic interventions, such as biliary stent placement, as potential risk factors for multidrug-resistant (MDR) infections—positioning our study not only as descriptive, but also exploratory in terms of resistance determinants.
- In Figure 2, the Y-axis is currently not labelled. Please revise the figure to indicate that the Y-axis represents the percentage (%) of total cases.
Response: Thank you for your comment. We have corrected the error.
- Please clarify the legends for Tables 5 and 7:
- For Table 7, specify whether the comparisons (p value) are between naïve papilla vs. stented common bile duct (CBD), between centers (CCH vs HLH), or both. Also indicate the statistical test(s) used.
- Similarly, table 7 does not specify the groups used for calculation of p- value.
Response: Thank you for your comment. The legends for Table 5 are as follows:
- N-CCH - Naïve papilla Colentina Hospital = patients with no previous history of endoscopic sphincterotomy admitted to the Colentina Clinical Hospital;
- N-HLH - Naïve papilla Haut-Lévêque Hospital = patients with no previous history of endoscopic sphincterotomy admitted to the Haut-Lévêque Hospital;
- S-CCH – Stented CBD Colentina Hospital = patients with a biliary stent already in place at the time of admission in Colentina Clinical Hospital
- S-HLH – Stented CBD Haut-Lévêque Hospital = patients with a biliary stent already in place at the time of admission in Haut-Lévêque Hospital
Regarding the former Table 7 (now Table 8 after adding the antibiotic use section), it represents a statistical method used to model the relationship between one or more independent variables (predictors), in our case represented by “Age > 75 years old“, “Male gender”, “Polymicrobial bile cultures”, “Positive blood culture” etc. and a binary dependent variable—that is, an outcome, in our case, the infection with a MDR microorganism. Unlike in a pairwise comparison using the Chi-square or Fisher's test, where the p-value show how significant is the difference between two groups, in this case, the p-value shows whether the association between the predictor and the outcome (dependent variable) is statistically significant. In our case, the p-value was calculated relative to the total number of patients in both medical units (Total CCH vs. Total HLH).
- Please correct- Candida spp. are fungi and not bacteria. Additionally, Candida spp. require specific methods for accurate identification, please specify whether the VITEK 2 YST card or another validated fungal identification system was used. This clarification is important to ensure the reliability of the data reported in Table 3 and Figure 3.
Response: Thank you for your comment. We have corrected these errors and added the additional information in the manuscript.
- Please ensure that all scientific names of microorganisms are italicized throughout the manuscript, following standard scientific style conventions.
Response: Thank you for your comment. We have corrected these errors.
Round 2
Reviewer 2 Report
Comments and Suggestions for Authors
The legends for Tables 5 and 8 (formerly Table 7) still do not adequately describe the statistical methods used. Specifically, the legends should clearly state which statistical test(s) were applied to generate the p-values and specify the comparison groups (e.g., naïve vs. stented CBD, CCH vs. HLH, or both). Additionally, the legends should be self-contained and independently describe the table contents without requiring the reader to refer back to the main text. This is essential for clarity and accurate interpretation of the results.
Author Response
The legends for Tables 5 and 8 (formerly Table 7) still do not adequately describe the statistical methods used. Specifically, the legends should clearly state which statistical test(s) were applied to generate the p-values and specify the comparison groups (e.g., naïve vs. stented CBD, CCH vs. HLH, or both). Additionally, the legends should be self-contained and independently describe the table contents without requiring the reader to refer back to the main text. This is essential for clarity and accurate interpretation of the results.
Response:
Dear Peer-Reviewer,
The legends of the two aforementioned tables were clarified following your suggestions. We thank you for your feedback and hope that the manuscript is now suitable for publication.